# Thrombo-Inflammation in COVID-19 and Sickle Cell Disease: Two Faces of the Same Coin

**DOI:** 10.3390/biomedicines11020338

**Published:** 2023-01-25

**Authors:** Kate Chander Chiang, Ajay Gupta, Prithu Sundd, Lakshmanan Krishnamurti

**Affiliations:** 1KARE Biosciences, Orange, CA 89128, USA; 2Division of Nephrology, Hypertension and Kidney Transplantation, Department of Medicine, University of California Irvine (UCI) School of Medicine, Irvine, CA 92868, USA; 3Vascular Medicine Institute and Division of Pulmonary, Allergy and Critical Care Medicine, University of Pittsburgh School of Medicine, Pittsburgh, PA 15261, USA; 4Division of Pediatric Hematology-Oncology, Yale School of Medicine, New Haven, CT 06510, USA

**Keywords:** sickle cell disease, COVID-19, SARS-CoV-2, vaso-occlusive crisis, pain, thromboxane, prostaglandin D_2_, thrombo-inflammation, acute chest syndrome, ramatroban

## Abstract

People with sickle cell disease (SCD) are at greater risk of severe illness and death from respiratory infections, including COVID-19, than people without SCD (Centers for Disease Control and Prevention, USA). Vaso-occlusive crises (VOC) in SCD and severe SARS-CoV-2 infection are both characterized by thrombo-inflammation mediated by endothelial injury, complement activation, inflammatory lipid storm, platelet activation, platelet-leukocyte adhesion, and activation of the coagulation cascade. Notably, lipid mediators, including thromboxane A_2_, significantly increase in severe COVID-19 and SCD. In addition, the release of thromboxane A_2_ from endothelial cells and macrophages stimulates platelets to release microvesicles, which are harbingers of multicellular adhesion and thrombo-inflammation. Currently, there are limited therapeutic strategies targeting platelet-neutrophil activation and thrombo-inflammation in either SCD or COVID-19 during acute crisis. However, due to many similarities between the pathobiology of thrombo-inflammation in SCD and COVID-19, therapies targeting one disease may likely be effective in the other. Therefore, the preclinical and clinical research spurred by the COVID-19 pandemic, including clinical trials of anti-thrombotic agents, are potentially applicable to VOC. Here, we first outline the parallels between SCD and COVID-19; second, review the role of lipid mediators in the pathogenesis of these diseases; and lastly, examine the therapeutic targets and potential treatments for the two diseases.

## 1. Introduction

During the current COVID-19 pandemic, over 550 million people have been infected with the SARS-CoV-2 virus, and more than 6 million people have died. Investigators have reported clinical outcomes of SCD patients who developed COVID-19 during the current pandemic [1,2,3,4,5,6]. Some studies have demonstrated a more effective antiviral response against SARS-CoV-2 in patients with SCD, leading to a lower incidence of COVID-19 complications [7,8,9]. However, most studies have reported worse outcomes with COVID-19 in SCD patients compared to the general population. Among 178 persons with SCD in the United States who were reported to an SCD-coronavirus case registry, 122 (69%) were hospitalized, and 13 (7%) died [2]. A study based on electronic health record data from a multisite research network compared outcomes of African Americans with COVID-19 with or without SCD or heterozygous states of sickle cell trait (SCT) [1]. After 1:1 propensity score matching (based on age, sex, and other preexisting comorbidities), patients with COVID-19 and SCD remained at a higher risk of hospitalization (relative risk [RR], 2.0; 95% CI, 1.5–2.7) and development of pneumonia (RR, 2.4; 95% CI, 1.6–3.4) and pain (RR, 3.4; 95% CI, 2.5–4.8), compared with African Americans without SCD or SCT.

In a prospective study of 3500 pediatric and adult patients with SCD treated at 5 academic centers in the U.S., 66 patients developed COVID-19 [3]. During a follow-up period of 3 months after diagnosis of SARS-CoV-2 infection, 75% of patients were hospitalized, and the mortality rate was 10.6%. Vaso-occlusive pain was the most common presenting symptom. Acute chest syndrome occurred in 60% of hospitalized patients and all patients with a fatal outcome. Older age and a history of pulmonary hypertension, congestive heart failure, chronic kidney disease, and stroke were risk factors for mortality. Laboratory parameters in those who died included higher creatinine, lactate dehydrogenase, and D-dimer levels. In hospitalized patients, anticoagulant use was twice more common in patients who survived. Furthermore, all mortality occurred in patients not on disease-modifying therapy for SCD.

In a genetic association study of 2729 persons with sickle cell trait (SCT) and 129,848 who were SCT-negative, individuals with SCT had a number of preexisting kidney conditions that were associated with unfavorable outcomes following COVID-19 [10]. The presence of SCT was associated with increased risk of mortality and acute kidney failure following COVID-19, suggesting that SCT is also a prognostic factor for COVID-19 [10].

Thus, SCD has, in fact, emerged as one of the most important comorbidities conferring a high risk of mortality from COVID 19 that far exceeds the risk associated with chronic kidney disease, leukemias and lymphomas, heart failure, diabetes, obesity, lung cancer, acute myocardial infarction, chronic obstructive pulmonary disease, tobacco use, ischemic heart disease, and hypertension in the most extensive comorbidity analysis of COVID-19 patients to date [11]. Other adverse outcomes may also include SCD-related chronic organ dysfunction, such as chronic persistent pain, lung, and kidney injury; fragmented care; poor access to quality care; and interruptions in care as a result of fear of exposure to COVID.

These case reports, case series, and registry-based cohorts provide evidence of a high risk of severe clinical course in SCD patients with COVID-19 and suggest an interaction between sickle cell and COVID-19 pathophysiology, while providing critical insights that may help generate mechanistic hypotheses and design prospective clinical trials.

It has been proposed that SCD is associated with impaired oxygen exchange, which may be further impeded during the inflammatory phase of COVID-19. However, complications, such as cerebrovascular events in SCD patients with COVID-19, have not been reported. Therefore, we postulate that endothelial injury, thrombo-inflammation, microvascular thrombosis, and resulting vaso-occlusive disease in SCD may be amplified by similar processes initiated by the SARS-CoV-2 virus and vice versa, adding to the risk of morbidity and mortality from any single disease. In this review, we examine the two diseases’ pathobiological processes and tease out the common pathways that may present a therapeutic target, potentially benefiting thousands of SCD patients worldwide during the COVID-19 pandemic.

## 2. Pathophysiology of Sickle Cell Disease and COVID-19

SCD affects millions of children and adults globally, including about 100,000 in the United States [12]. The average life expectancy of an SCD patient at birth is 42–47 years in the United States [13], compared to about 79 years for the general U.S. population. SCD has a profound adverse impact on the quality of life. The current therapeutic options for SCD include hydration, blood transfusions, hydroxyurea, L-glutamine, crizanlizumab, and voxelotor.

The clinical hallmark of SCD is vaso-occlusive crises (VOCs), also referred to as pain crises. Cerebral vasculopathy, in its most devastating form, results in arterial thrombosis, which leads to cerebral infarction and stroke in early childhood [14,15]. SCD remains one of the most common causes of stroke in children [16]. The risk of stroke is higher during the first decade and is more significant between ages 2 and 5, when it reaches about 1% per year [17]. About 10% of SCD patients have a clinically apparent stroke before the age of 20, and the risk increases to about 25% by the age of 45 years [17]. Thrombotic vasculopathy in SCD is accompanied by significant organ dysfunction, morbidity diminished quality of life, and premature mortality [18,19]. Despite recent therapeutic advances, SCD patients remain at a high risk of developing VOCs and vascular complications. Thus, it is imperative to address the scientific gaps in our understanding of the mechanisms underlying the thrombo-inflammatory state, which is generally characterized by sickle RBC-mediated endothelial inflammation/dysfunction and coagulation activation leading to vessel injury, leakage, and vascular thrombosis [15].

The SARS-CoV-2 infection causes COVID-19 disease, which in its severe form can present with thrombotic microangiopathy, pulmonary thrombosis, pedal acro-ischemia (“COVID-toes”), arterial clots, strokes, cardiomyopathy, coronary and systemic vasculitis, bleeding, deep venous thrombosis, pulmonary embolism, and microvascular thrombosis in renal, cardiac, and brain vasculature [20,21,22,23,24,25]. Furthermore, necropsies have revealed inflammatory microvascular thrombi containing neutrophils, platelets, and neutrophil extracellular traps (NETs) in the pulmonary, hepatic, renal, and cardiac microvasculature as the hallmark of severe COVID-19 disease and the underlying cause of multi-organ failure [24,26,27]. Similar thrombo-inflammatory processes mediated by cell-free hemoglobin have been observed and proposed in SCD, with emphasis on platelet activation [15,28], a key driver of thrombo-inflammation in COVID-19 [24]. Consistent with the above, COVID-19 has been associated with increased risk of thrombosis in patients with SCD [29]. Therefore, we propose that the emerging therapies targeting platelet-mediated thrombo-inflammation in COVID-19 may serve as potential therapies for VOC.

In SCD, VOC often causes acute chest syndrome (ACS), defined as the presence of fever and/or new respiratory symptoms, accompanied by a new pulmonary infiltrate on a chest X-ray [30]. This is very similar to acute pneumonia in COVID-19. However, there are significant differences in the clinical presentation and underlying pathological basis for thrombo-inflammation in COVID-19 versus SCD. The incidence rate of ACS is highest at 2 to 4 years of age among children with SCD, with a rate of 25.3 per 100 patient-years, and decreases to 8.87 per 100 patient-years in adults >20 years of age with HbSS [31]. On the other hand, acute pneumonia and respiratory failure are more common in adults with COVID-19 [32].

Pulmonary complications associated with COVID-19 or SCD reveal similar underlying pathobiology and therapeutic targets. Respiratory distress in COVID-19 occurs in part due to pulmonary platelet microvascular thrombosis [20]. However, the triggering events of ACS in SCD patients may vary. Although conventional wisdom suggests ACS occurs secondary to fat embolism in SCD, more recent evidence from CT studies has demonstrated in situ pulmonary thrombosis in 10–20% of ACS patients [33,34,35,36,37,38,39,40]. Interestingly, in 16% of 538 SCD patients with ACS, pulmonary infarction or thrombosis were the triggering events rather than infection or fat embolism [41]. Still, fat embolism due to bone marrow infarction occurs in about 40% of both children [42] and adults [43,44,45], and leads to disseminated pulmonary platelet thrombi with a sharp and significant decline in platelet count prior to death [46]. It has been proposed that platelet inhibition at steady state or in the hemodynamically stable acute crisis might be an important therapeutic addition to prevent the progression of ACS in SCD [46]. A comparative analysis of the pathobiology of SCD and COVID-19, particularly in the context of endothelial injury, platelet activation, and multicellular adhesion, may help to identify potential therapies for the thrombo-inflammation in both diseases.

## 3. Mechanisms of Multicellular Adhesion and Thrombo-Inflammation in Sickle Cell Disease and COVID-19

Complications associated with thrombo-inflammation in SCD have uncanny similarities to those in COVID-19. In SCD, injury to the red blood cell (RBC) membrane mediates endothelial damage and inflammation, leading to multi-organ vasculopathy [47]. Hemoglobin S polymerization impairs the deformability of the RBC and causes oxidative injury and destruction of the RBC [47]. RBC injury exposes phosphatidyl serine and releases Hb and other intracellular contents [47]. This, in turn, depletes NO, increases endothelial adherence, releases pro-inflammatory cytokines, and activates coagulation, causing ischemia, reperfusion injury, and vascular damage [47,48,49,50,51]. Similar inflammatory processes observed during SARS-CoV-2 infection are evidenced by the elevated expression of leukocyte adhesion molecules in the pulmonary vasculature [20] and the presence of a proinflammatory lipid/thromboxane storm [52].

### 3.1. Endothelial Cell Injury and Activation: Role in Thrombo-Inflammation in SCD and COVID-19

Endothelial cell injury and activation lie at the heart of the prothrombotic state in both SCD and COVID-19 (Figure 1). Vascular endothelium is activated in SCD, regardless of the patient’s clinical status, with markedly increased expression of adhesion molecules, including intercellular adhesion molecule 1 (ICAM-1), vascular-cell adhesion molecule 1 (VCAM-1), E-selectin, and P-selectin [53]. SARS-CoV-2 virus directly infects and damages the endothelial cells, which initiates a cascade of events, leading to intussusceptive angiogenesis and microvascular thrombosis [20]. SCD and COVID-19 are characterized by interactions among activated endothelial cells, platelets, and leukocytes, leading to thrombo-inflammation and vascular occlusion [54]. Most notably, endothelial inflammation induces surface expression of adhesion molecules, including P-selectin, and release of prothrombotic granule contents (von Willebrand factor and FVIII), both effects enhancing leukocyte/platelet adhesion [15]. Intravascular release of tissue factor (TF) also contributes to the polarization toward a prothrombotic state [23,55,56,57]. Release of cell-free heme activates converging inflammatory pathways, such as TLR4 signaling [58], formation of neutrophil extracellular traps (NETs) [24,59], and priming of the inflammasome (NLRP3) pathway, leading to the release of interleukin-1β (IL-1β) and IL-18 by leukocytes, platelets, and endothelial cells, which contributes to the development of a sterile thrombo-inflammatory state in SCD [28,51,60,61].

### 3.2. Platelet Activation: Role in Thrombo-Inflammation in SCD and COVID-19

The pathogenesis of platelet activation in COVID-19 and SCD is multifactorial. COVID-19 and SCD activate platelets by association with the SARS-CoV-2 virus or direct activation with cell-free hemoglobin. Sickling and vaso-occlusion in SCD lead to hemolysis and subsequent release of cell-free hemoglobin [15]. Free plasma hemoglobin generates reactive oxygen species, a potent nitric oxide scavenger [62]. Nitric oxide scavenging promotes platelet activation and endothelial dysfunction [62]. Under physiological conditions, free heme is scavenged by the plasma protein hemopexin and is subsequently catabolized by heme oxygenase-1 into carbon monoxide, biliverdin, and ferrous iron (Fe^2+^) [63]. Acute or chronic hemolysis exhausts this scavenging system for heme, leading to an increase in free heme in the blood [63]. Upon release, reduced heme is rapidly and spontaneously oxidized in the blood into its ferric (Fe^3+^) form, hemin, with increased levels observed in hemolytic diseases [63]. Hemin has been implicated in the pathogenesis of ACS, one of the leading causes of death in SCD [64]. Hemin activates platelets as a ligand for C-type-lectin-like receptor 2 (CLEC2) [63]. Hemin-induced aggregation of human platelets is abolished by pre-incubation of hemin with a recombinant dimeric form of CLEC2 [63]. This indicates a role for platelet CLEC2 in sickle cell-mediated platelet activation (Figure 2). Cell-free heme also amplifies inflammation [65] by activating inflammatory pathways, including TLR signaling [66], gasdermin D-dependent NET formation [59,61], platelet-inflammasome activation, and generation of IL-1β-carrying platelet extracellular vesicles and priming of the inflammasome, leading to platelet-neutrophil aggregation and vaso-occlusion [28,51]. Consistent with the above, incubation of human peripheral neutrophils with VOC plasma produced significantly more NETs compared to non-sickle and steady state plasma [67]. NET generation in SCD is caused by sterile inflammation [61]. Additionally, during SCD-induced bone marrow infarction, the bone marrow undergoes stress reticulocytosis. As a result, it releases immature erythrocyte or reticulocytes [62] with surface expression of adhesion molecules, such as CD36 and α4β1 integrin [51,68], which contribute to platelet activation and thrombo-inflammation.

SARS-CoV-2 viral hemagglutinins can bind to circulating red blood cells (RBCs) and induce agglutination and clumping of RBCs [69]. First, SARS-CoV-2 binds to RBCs in vitro [70] and clinically in COVID-19 patients [69,71]. Second, although fusion and replication of SARS-CoV-2 occur via ACE2, such hemagglutinating viruses initially attach to infective targets and clump with blood cells via much more abundantly distributed sialic acid glycoconjugate binding sites [69]. SARS-CoV-2, in particular, binds to these sialic acid sites [69]. Third, certain enveloped viruses express an enzyme, hemagglutinin esterase, that counteracts viral-RBC clumping, but is lacking in the SARS-CoV-2 virus [69]. These hemagglutinating properties of SARS-CoV-2 establish a framework for “catch and clump” induction of microvascular occlusion [69].

Subsequent hemolysis marked by elevated levels of LDH and thrombotic microangiopathy may play a role in platelet activation in COVID-19. Despite only minimal symptoms of COVID-19, 13 of the 34 children studied had thrombotic microangiopathy concurrent with complement activation marked by increased plasma sC5b-9 levels [72]. Furthermore, in 181 adults hospitalized for COVID-19, an increased percentage of schistocytes were correlated with decreased platelet count and increased markers of hemolysis, such as LDH [73]. The percentage of schistocytes was higher in those who died than those who survived [73]. Thus, thrombotic microangiopathy plays a significant role in platelet activation and morbidity in COVID-19, potentially through the release of free heme.

D-dimers are a prognostic marker of COVID-19 [74]. D-dimer levels are more likely to be abnormal in severely and critically ill patients, compared with mild and ordinary cases. At the same time, D-dimer levels of patients who died are significantly higher than those of surviving patients [74]. D-dimer levels are also raised in VOC and in most SCD patients with an abnormal chest X-ray (Table 1) [75] indicative of a prothrombotic state.

Platelet activation plays a crucial role in both SCD and COVID-19. Platelet-derived microparticles are a biomarker of vaso-occlusive events in severe cases of SCD, while erythrocyte-derived microparticles are higher in non-severe disease [79]. Platelet extracellular vesicles and markers of platelet degranulation, including platelet factor 4 and serotonin in the blood, are also increased in COVID-19 [80].

In SCD, platelet activation and release of microparticles is likely mediated by heme-induced platelet CLEC2 receptor or NLRP3 inflammasome activation [28,63]. However, in COVID-19, heme-induced platelet CLEC2 activation has not been reported, to the best of our knowledge [81]. On the other hand, SARS-CoV-2 associates with platelets [80] possibly by binding of SARS-CoV-2 spike receptor binding domain (S-RBD) to dendritic cell-specific intercellular adhesion molecule-3-grabbing non-integrin (DC-SIGN) and liver/lymph node-specific intracellular adhesions molecule-3-grabbing integrin (L-SIGN) [82]. This is reminiscent of dengue virus-induced activation of platelets by binding to a DC-SIGN/CLEC2 hetero-multivalent receptor complex, resulting in CLEC2 activation and platelet degranulation with the release of extracellular vesicles, including exosomes and microvesicles [83].

Upon activation, the CLEC2 receptor undergoes tyrosine phosphorylation mediated by thromboxane A_2_ (TxA_2_) [84]. This leads to downstream phosphorylation of spleen tyrosine kinase and phospholipase γ2, potentiated by TxA_2_ [84]. This cooperation between CLEC2 and TxA_2_ signaling is critical for platelet activation (Figure 2) [84]. Platelet activation leads to the release of exosomes and microvesicles that further activate CLEC5A and TLR2 receptors on neutrophils and macrophages, thereby inducing NET formation and proinflammatory cytokine release [83]. Therefore, CLEC2 signaling is a potential therapeutic target in both SCD and COVID-19 (Figure 2).

### 3.3. P-Selectin: Role in Thrombo-Inflammation in SCD and COVID-19

Upregulation of P-selectin in endothelial cells and platelets contributes to the cell–cell interactions involved in vaso-occlusion and sickle cell-related pain crisis [85,86], and plasma levels of soluble P-selectin are markedly increased in vaso-occlusive SCD [87]. P-selectin is a well-recognized therapeutic target in SCD, and its inhibition by crizanlizumab, a humanized monoclonal antibody, significantly lowers rates of sickle cell-related pain crises [85]. Similarly, plasma levels of soluble P-selectin are markedly increased in COVID-19 [88]. Platelet P-selectin surface expression is upregulated in COVID-19 and positively correlates with platelet-monocyte aggregates in infected subjects [23]. In a randomized, placebo, controlled clinical trial amongst 54 hospitalized COVID-19 patients, crizanlizumab reduced P-selectin levels by 89% while promoting thrombolysis, as suggested by a 77% increase in D-dimers and decreased prothrombin fragments, but there was no difference in the clinical outcomes (the CRITICAL study) [89].

### 3.4. Tissue Factor: Role in Thrombo-Inflammation in SCD and COVID-19

TF is a transmembrane protein that functions as a high-affinity receptor for factors VII and VIIa and is the primary cellular initiator of blood coagulation during endothelial injury [90,91]. Formation of the TF-factor VIIa (FVIIa) complex leads to the activation of both FX and FIX, with subsequent thrombin generation, fibrin deposition, and activation of platelets [92]. Under normal conditions, endothelial cells and blood cells, such as monocytes, do not express TF [55]. On the other hand, total circulating microparticles expressing TF, mainly derived from monocytes and endothelial cells, are elevated in sickle cell crisis, compared to steady state and healthy controls [55]. Interestingly, TF inhibition in transgenic SCD mice significantly attenuates heme-induced microvascular stasis and prevents lung vaso-occlusion mediated by arteriolar neutrophil-platelet microemboli [93]. In severe COVID-19, platelet activation and TF expression by monocytes leading to platelet-monocyte interaction are associated with COVID-19 severity and mortality [23].

### 3.5. CD40L: Role in Thrombo-Inflammation in SCD and COVID-19

CD40L is a type II transmembrane protein expressed primarily by activated T cells, activated B cells, and platelets, and under inflammatory conditions, it is also induced on monocytic cells, natural killer cells, mast cells, and basophils [94,95]. CD40L binds to CD40 expressed on a variety of cells, including dendritic cells, monocytes, platelet, and macrophages [95]. CD40L/CD40 interactions are pivotal in different cellular immune processes [95]. Notably, platelets release CD40L, which contributes to chronic inflammation in SCD [96]. Elevated levels of circulating CD40L have been associated with acute chest syndrome (ACS) in SCD [96,97]. Platelets from SARS-CoV-2 patients are also more prone to release of soluble CD40L upon exposure to thrombin, compared to healthy controls [80].

Activated platelets are also a significant source of thrombospondin-1, another protein related to the incidence of ACS and vaso-occlusive episodes [98]. However, thrombospondin has not been examined in COVID-19, to the best of our knowledge.

### 3.6. NLRP3 Inflammasome: Role in Thromboinflammation in SCD and COVID-19

Platelets are known to play a role in the detection and regulation of infection [99]. Viruses, such as the dengue virus, lead to platelet activation [99]. Platelets sense pathogens and host damage through recognition of pathogen-associated molecular patterns or damage-associated molecular patterns (DAMPs) using receptors [99]. C-type lectin receptors DC-SIGN and CLEC2 are involved in the binding of different viruses, as well as the recognition of DAMPs, such as hemin and mitochondrial DNA [99]. Platelets are highly activated in COVID-19. They are likely involved in boosting the inflammasome capacity of innate immune cells, including human macrophages and neutrophils, and IL-1 production by monocytes [24,100,101,102]. An unknown platelet-derived soluble factor enhances NLRP3 transcription and inflammasome activation [102]. We postulate that CLEC2-induced platelet activation leads to the release of exosomes and microvesicles, which stimulate the CLEC5A and TLR2 receptors on innate immune cells, leading to NLRP3 inflammasome activation and pyroptosis [83].

SARS-CoV-2 virus also induces inflammasome activation and cell death by pyroptosis in human monocytes, hematopoietic stem/progenitor cells, and endothelial progenitor cells [103,104]. Pyroptosis was dependent on caspase-1 engagement, before IL-1ß production and inflammatory cell death [103]. Furthermore, examination of the whole blood transcriptome in COVID-19 patients has revealed that the dysregulated immune system in COVID-19 is characterized by highly specific neutrophil activation-associated signatures [105], with an increase in immature neutrophils with NLRP3 inflammasome activation [101].

NLRP3 inflammasome is also upregulated in SCD patients under steady state conditions, compared with healthy controls, and is further upregulated when patients experience an acute pain crisis [106]. Platelet-inflammasome activation led to the generation of IL-1β and caspase-1-carrying platelet extracellular vesicles that bind to neutrophils and promote platelet-neutrophil aggregation in lung arterioles of SCD mice in vivo and SCD human blood in microfluidics in vitro [28]. Inhibition of the inflammasome effector caspase-1 or IL-1β pathway attenuated platelet extracellular vesicle generation, prevented platelet-neutrophil aggregation, and restored microvascular blood flow [28]. More recent findings show that sterile inflammation in SCD promotes caspase-11/4-dependent activation (cleavage) of pyroptotic effector gasdermin-D (GSDMD) in neutrophils, which leads to generation of NETs in the liver [61]. These NETs embolize from the liver to the lung to promote P-selectin-independent lung vaso-occlusion in SCD [61]. Interestingly, GSDMD is highly expressed on the BALF and blood neutrophils of COVID-19 patients [107]. Image analysis of lung autopsies of patients who died from COVID-19 revealed the presence of NET structures associated with activated GSDMD-NT fraction [107]. In cell cultures of neutrophils from COVID-19 patients, disulfiram, a GSDMD inhibitor, inhibited release of NETs in a concentration-dependent manner [107]. Therefore, in both SCD and COVID-19, activation of inflammasome in platelets, monocytes, and neutrophils and GSDMD-dependent NETosis play a key role in initiating inflammation and tissue injury (Figure 2).

### 3.7. Nitric Oxide: Role in Thrombo-Inflammation in SCD and COVID-19

Both COVID-19 and SCD are associated with endothelial injury and activation. Following endothelial injury, nitric oxide (NO) has been shown to serve many vasoprotective roles, including inhibition of platelet aggregation and adherence to the site of injury, inhibition of leukocyte adherence, inhibition of vascular smooth muscle cell proliferation and migration, and stimulation of endothelial cell growth [51,62,108,109,110].

In SCD, cell-free plasma hemoglobin resulting from intravascular hemolysis consumes NO very rapidly [111], dramatically limiting NO bioavailability [112,113]. Inhaled NO has shown evidence of efficacy in mouse models of SCD, but in a phase II placebo-controlled trial of inhaled NO gas in SCD patients with VOC, NO did not improve the time to crisis resolution [114].

NO deficiency has also been observed among COVID-19 patients, and it may cause vascular smooth muscle contractions [115], reducing the ability to neutralize ROS and NO-mediated antiviral capability [116,117,118]. Nitric oxide has been widely proposed as a potential treatment for COVID-19 [119]. However, inhaled NO gas may be rapidly sequestered by superoxide, forming peroxynitrite, which is known to cause lung damage and cell death [120]. It is plausible that NO in SCD [114] and COVID-19 [121] could lack therapeutic benefit in an environment of oxidative stress or in the absence of sufficient L-arginine bioavailability [120].

### 3.8. TGFβ: Role in Thromboinflammation in SCD and COVID-19

The transforming growth factor (TGF-β) superfamily is composed of a large group of proteins that are fundamental in regulating various biological processes, such as extracellular deposition, cell differentiation and growth, tissue homeostasis and repair, and immune and inflammatory responses [122]. The TGF-β subfamily is a central mediator of fibrogenesis and a crucial regulator of fibroblast phenotype and function. There are three known isoforms of TGF-β expressed in mammalian tissue, including TGF-β1, 2, and 3. TGF-β1 is the most abundant and ubiquitously expressed isoform and is associated with the development of tissue fibrosis [122,123,124]. Various animal models support the role of TGF-β1 in mediating hepatic, renal, pulmonary, and cardiac fibrosis [125,126,127,128]. Interestingly, platelets contain 40 to 100 times more TGF-β1 than other cells and rapidly release TGF-β1 upon activation [129]. This is consistent with the positive correlation between plasma TGF-β1 and platelet and white blood cell counts in patients with steady state SCD [130,131]. Interestingly, early, untimely TGF-β responses in SARS-CoV-2 infection limit the antiviral function of natural killer (NK) cells [132]. Therefore, TGF-β has been proposed as a therapeutic target in both SCD and COVID-19 [130,132].

### 3.9. Lipoxygenase: Role in Thrombo-Inflammation in SCD and COVID-19

Activation of the lipoxygenase (LOX) pathway in SCD and COVID-19 promotes generation of arachidonic acid-derived eicosanoids. In isolated rat lungs, perfusion with HbSS peptide leads to more LOX metabolite LTC_4_, compared to HbAA-perfused lungs (10.40 ± 0.62 vs. 0.92 ± 0.22 ng/g dry lung weight (mean ± SEM; *p* = 0.0001)) [133]. Cysteinyl leukotrienes have been implicated in the pulmonary manifestations of SCD, including acute chest syndrome [133]. However, targeting of cysteinyl leukotriene receptor with montelukast did not improve pain, pulmonary function, or microvascular blood flow in a phase 2 randomized placebo-controlled trial in 42 adolescent/adult SCD patients [134].

LOX activity is increased in the plasma of COVID-19 patients, with higher activity in patients who survived [135]. LOX-derived leukotriene production is also significantly increased in the BALF of COVID-19 patients [136]. Montelukast has been shown to inhibit in vitro platelet activation induced by plasma from COVID-19 patients [137]. Montelukast prevented surface expression of tissue factor (TF) and P-selectin, and reduced circulating monocyte- and granulocyte-platelet aggregates and TF^+^-circulating microvesicles [137]. In a prospective randomized controlled study of montelukast in 180 hospitalized COVID-19 patients, patients receiving montelukast had significantly lower LDH, fibrinogen, D-dimer, CRP, and procalcitonin, compared to standard of care alone [138]. Additionally, treatment with montelukast significantly reduced the progression to macrophage activation syndrome and respiratory failure, while significantly reducing the length of hospital stay [138].

## 4. Thromboxane A_2_-A Key Mediator of Thrombo-inflammation by Regulation of Platelet Activation, NO Synthesis, and Expression of P-Selectin, CD40L, Tissue Factor, and TGF-β

Thromboxane A_2_ (TxA_2_), a key mediator of thrombosis, is released by platelets, endothelial cells, macrophages, and neutrophils [139]. TxA_2_ binds to the thromboxane-prostanoid (TP) receptor on platelets, thereby stimulating activation and aggregation of platelets [139]. Cooperation between TxA_2_/TP receptor and CLEC2 receptor signaling pathways is critical for CLEC2-induced platelet activation [84].

Thromboxane A_2_ generation is markedly stimulated in both SCD and COVID-19. In SCD, TxB_2_, and 2,3-dinor-TxB_2_, a terminal metabolite of TxB_2_, were significantly elevated in the urine and plasma of steady state SCD patients, compared to healthy HbAA controls (Table 2) [140,141]. Moreover, in isolated rat lungs co-perfused with sickle (HbSS) erythrocytes and platelet-rich plasma, TxA_2_ levels increased over 10-fold more than with normal (HbAA) erythrocytes [142]. In severe COVID-19 patients, bronchoalveolar lavage fluid presents a picture of an inflammatory lipid storm, with marked increases in fatty acid levels and a predominance of cyclooxygenase metabolites, notably, thromboxane B_2_ >> PGE_2_ > PGD_2_ [52]. Plasma levels of TxB_2_, a stable metabolite of TxA_2_, are also markedly increased in severe COVID-19 patients [23]. Interestingly, proinflammatory eicosanoids TxB_2_ and anti-inflammatory eicosanoids 15d-PDJ_2_ and 12HETE were elevated in the plasma of 66 COVID-19 survivors [135]. However, only the plasma levels of TxB_2_, and not 15d-PDJ_2_ or 12HETE, were elevated in 22 patients who died from COVID-19 [135]. Considering the marked increase of TxA_2_ in both SCD and COVID-19, we postulate the potential role of TxA_2_ in the pathogenesis of the proinflammatory state that contributes to the thrombo-inflammation observed in both diseases.

There is growing evidence that cyclooxygenase enzymes, COX-1 and COX-2, mediate the thromboxane generation underlying thrombo-inflammation in SCD and COVID-19 [81]. COX-2 is an inducible enzyme, while COX-1 is constitutive. COX-2 expression is stimulated by inflammation, a cardinal feature of both VOC and COVID-19. Endothelial COX-2 expression was markedly increased in transgenic BERK SCD mice [145]. SARS-CoV-2 infection of iPSC-derived cardiomyocyte cells led to >50-fold increase in COX-2 gene expression (Dr. S. T. Reddy, UCLA, personal communication, based on analysis of the supplemental material in reference [146]. In addition to upregulating COX-2 in living human lung slices, the SARS-CoV-2 virus reduces the prostaglandin-degrading enzyme 15-hydroxyprostaglandin-dehydrogenase [147]. TxA_2_ generation is COX-2 mediated in states of high TxA_2_ generation, such as in inflammation, infection, and obesity [148]. Low-dose aspirin is unable to inhibit COX-2; hence, aspirin resistance is common in states with COX-2-dependent TxA_2_ generation [149]. TxA_2_ generation by platelets and endothelial cells stimulates expression of P-selectin, ICAM-1, and VCAM-1 on endothelial cells and release of tissue factor [150,151,152].

### 4.1. Thromboxane A_2_-Mediated P-Selectin Expression

TxA_2_ plays a role in the platelet expression of P-selectin. It was demonstrated that the percentage of P-selectin-positive platelets in TP receptor knockout mice on day 1 was significantly reduced, compared with that in wild-type mice [151]. Therefore, TxA_2_ blockade may be another effective method to target P-selectin without the need for IV administration of anti-P-selectin Ab [85].

### 4.2. Thromboxane A_2_-Mediated Tissue Factor Expression

TxA_2_ has been shown to mediate TF expression on endothelial cells and monocytes [152]. TP receptor agonism induced TF expression in endothelial cells. In contrast, a TP receptor antagonist reduced endothelial expression of TF after TNF-α induction [152,153]. Similarly, lipopolysaccharide-induced TF expression on human monocytes was abrogated by a TP receptor antagonist [154].

### 4.3. Thromboxane A_2_-Mediated CD40L Expression

Plasma CD40L has been correlated with urinary 11-dehydro-TxB_2_, a stable metabolite of TxA_2_ in diabetic patients [155]. Upon treatment with low dose aspirin (100 mg/day), plasma CD40L decreased, along with a reduction in urinary 11-dehydro-TxB_2_ and whole blood TxB_2_ production [155]. Patients with higher excretion of 11-dehydro-TxB_2_ had increased levels of CD40L [155]. Therefore, targeting TxA_2_ may reduce the release of CD40L, potentially preventing ACS and vaso-occlusion in SCD.

### 4.4. Thromboxane A_2_-Induced Suppression of NO Synthesis

NO inhibits platelet activation via phosphorylation of the thromboxane prostanoid (TP) receptor [156]. In both vascular smooth muscle cells and platelets, the vasodilatory and platelet inhibitory effects of NO are known to be mediated by cGMP, which inhibits phospholipase C activation, inositol 1,4,5-triphosphate generation, and [Ca^2+^]_i_ mobilization [156]. NO stimulates cGMP production and activates cGMP-dependent protein kinase or G kinase [156]. TxA_2_ also directly inhibits nitric oxide synthase [157]. Nitrite accumulation was enhanced by TP receptor antagonists, seratrodast or ramatroban, in a model of IL-1β-stimulated rat aortic smooth muscle cells [157]. Therefore, TxA_2_ may play a role in NO deficiency in SCD, which would be alleviated by TP receptor blockade.

### 4.5. Thromboxane A_2_-Induced TGF-β Release

TxA_2_/TP receptor signaling stimulates activation of the TGF-β pathway [158,159]. Hypertensive PGI_2_ receptor knockout mice fed a high-salt diet exhibited elevated urinary TxA_2_ metabolites and left ventricular TP receptor overexpression, which accompanied cardiac collagen deposition and profibrotic TGF-β gene expression [160]. Inhibition of TxA_2_ biosynthesis with low-dose aspirin mitigated the increase in blood pressure, cardiac fibrosis, and left ventricular TGF-β gene expression [160]. Moreover, the number of myofibroblasts and extravasated platelets in the heart were also reduced [160]. This is consistent with TxA_2_-induced TGF-β gene expression in myofibroblasts [160].

## 5. Thromboxane A_2_ in Post-Capillary Venoconstriction in SCD and COVID-19

### 5.1. Post-Capillary Pulmonary Venous Constriction

Cardiopulmonary complications are the leading cause of death in patients with SCD, primarily resulting from diastolic heart failure (HF) and/or pulmonary hypertension (PH) [161]. From a hemodynamic standpoint, almost half of cases of SCD pulmonary hypertension reported in the literature have postcapillary or venous pulmonary hypertension [162]. Interestingly, U-46619, a TxA_2_ mimetic in a concentration of 1 nM, is sufficient to reduce the guinea pig pulmonary venous luminal area by 50% [163]. A 50% reduction in luminal area increases vascular resistance by 4-fold, indicating that sub-nanomolar concentrations of thromboxane A_2_ could produce meaningful increases in pulmonary venous resistance [163]. This is consistent with the measured effect of selective TP receptor antagonism in reducing pulmonary venous resistance and capillary pressure in patients with acute lung injury [164]. Moreover, TP receptor antagonism prevented right ventricular fibrosis and arrhythmias in a mouse model of pulmonary arterial hypertension [165]. Therefore, TxA_2_ released from platelets and pulmonary venous endothelial cells may cause: first, pulmonary venous hypertension and, second, left ventricular fibrosis secondary to elevated TGF-β levels and diastolic dysfunction.

TP receptor antagonism has also been shown to attenuate airway mucus hyperproduction induced by cigarette smoke [166] and reduce tissue edema in mouse models of acute lung injury [167]. In COVID-19, TP receptor blockade may rapidly reduce pulmonary capillary pressures, improve ventilation-perfusion matching, promote resolution of edema, reduce bronchoconstriction and airway mucus hyperproduction, improve lung compliance and gas exchange, and thereby mitigate respiratory distress and hypoxemia [168].

### 5.2. Post-Capillary Efferent Arteriole Constriction in Kidney Injury

Glomerular involvement is one of the most prominent renal manifestations observed in SCD. It is characterized by an early increase in glomerular filtration rate (GFR) associated with micro- or macroalbuminuria, followed by a gradual decline in GFR and chronic renal failure [169,170]. This is consistent with underlying vasculopathy in sickle cell nephropathy associated with cortical hyperperfusion, medullary hypoperfusion, and an increased stress-induced vasoconstrictive response [169]. Renal involvement is usually more severe in homozygous than heterozygous SCD. It contributes to diminished life expectancy and 16–18% of mortality in patients with SCD [169,171,172]. Acute kidney injury is emerging as a common and important sequelae of COVID-19, with rates as high as 33–43% in hospitalized patients [173,174,175,176]. In a prospective cohort study of 701 COVID-19 patients, proteinuria was reported in 43.9% of patients on admission to the hospital [177]. However, the involvement of hyperfiltration in COVID-19-associated kidney injury remains to be elicited.

Glomerular hyperfiltration is caused by either a net reduction of afferent (pre-capillary) arteriolar resistance or a net increase in efferent arteriolar (post-capillary) arteriolar resistance [178]. Thromboxane synthase inhibition or TP receptor antagonism in untreated streptozotocin-induced diabetic rats was shown to decrease renal blood flow, increase renal vascular resistance, and ameliorate renal hyperperfusion [179]. This is consistent with reduced microalbuminuria in diabetic patients treated with a TxA_2_ synthase inhibitor, likely due to a vasodilating effect predominantly exerted on the efferent arteriole [180,181]. However, this contrasts with in vitro findings that treatment of isolated perfused hydronephrotic rat kidney with TxA_2_ mimetic leads to preferential constriction of afferent arterioles [182]. Therefore, further studies are needed to clarify the role of TxA_2_ in post-capillary efferent arteriole constriction and glomerular involvement associated with vascular diseases, including SCD and COVID-19.

## 6. Complement Activation as an Inducer of Thrombo-Inflammation in SCD and COVID-19

The complement system is a critical innate immune defense against infections and an important driver of inflammation [183]. Activated complement can produce direct effector functions by target opsonization with cleaved complement component 3 (C3) and C4 fragments, promotion of inflammation with C3a and C5a, and direct cell lysis with the assembly of MAC C5b-9 complex [183]. Plasma concentrations of sC5b-9 are elevated in steady state SCD patients [184]. Complement deposition is also increased in cultured human endothelial cells incubated with SCD serum [184]. In SCD, levels of IL-1α were significantly higher in those with a history of acute splenic sequestration, a common feature of homozygous SCD, compared with matched normal controls [185]. The effect of complement activation on IL-1α and COX-2/TxA_2_ axes has been studied by treating porcine endothelial cells with human plasma containing xenoreactive antibodies and complement [186]. In this model, there is markedly increased expression of IL-1α, COX-2, and thromboxane synthase, leading to the generation of TxA_2_. The role of IL-1α in mediating the effect of complement activation was confirmed by the addition of an IL-1 receptor antagonist to the human serum, which prevented the release of PGE_2_ and TxA_2_ [186]. Therefore, complement activation can induce a prothrombotic state via the expression of IL-1α and COX-2, leading to generation of TxA_2_ (Figure 1).

Complement activation is also thought to play a critical role in immune-thrombosis and end-organ damage in COVID-19 [25]. Nucleocapsid protein of SARS-CoV-2 virus binds to the Mannan-binding, lectin-associated serine protease-2 (MASP-2), the lectin pathway’s effector enzyme, resulting in complement activation [187,188]. Lung tissue from deceased COVID-19 patients showed components of the lectin and terminal complement pathways, specifically MASP-2, complement factor 4d (C4d), and C5b-9 (i.e., the membrane attack complex) [187,188]. Activation of C4d (classical lectin pathway) and sC5b-9 (membrane attack complex) are also associated with respiratory failure in hospitalized adults with COVID-19 [189]. Furthermore, in children with COVID-19, there is evidence of complement activation with an increase in plasma sC5b-9 levels, even with only minimal symptoms of COVID-19, and 13 of the 34 children had thrombotic microangiopathy [72]. Most importantly, IL-1α is also released from necrotic and pyroptotic cells, including pneumocytes and endothelial cells, the primary site of an attack by SARS-CoV-2 [190].

## 7. Thromboxane A_2_ Is Enzymatically Converted into 11-Dehydro-Thromboxane A_2_, a Full Agonist of the Prostaglandin D_2_/DP2 Receptor Leading to Fibrosis and Inflammation

TxA_2_ is short-lived and rapidly transformed nonenzymatically in an aqueous solution to TxB_2_. TxB_2_ is further metabolized enzymatically to a series of compounds, of which 11-dehydro-TxB_2_ (11dhTxB_2_) is the major product found both in plasma and urine [191]. The dehydrogenase catalyzing the formation of 11dhTxB_2_ was tissue bound, with the highest activity in the lung [192]. Urinary excretion of 11dhTxB_2_ was markedly increased in recently hospitalized patients with COVID-19 and was predictive of plasma D-dimer levels, renal ischemia, the need for mechanical ventilation, and mortality [143]. Urinary 11dhTxB_2_ is also significantly elevated in SCD compared to healthy controls [141]. Interestingly, 11-dehydro-TxB_2_ is a full agonist of the D-prostanoid receptor 2 (DP2) for prostaglandin D_2_ (PGD_2_) [191].

PGD_2_/DP2 receptor signaling is known to mediate Th2 immune responses that are classically directed against extracellular non-phagocytosable pathogens, for instance, helminths [193,194,195]. The effectors for PGD_2_/DP2 receptor-mediated Th2 immune response are eosinophils, basophils, and mastocytes, as well as B cells (humoral immunity), and these are consistently elevated in COVID-19 [196]. IL-13, a type 2 cytokine released during PGD_2_/DP2 receptor signaling, increases hyaluronan accumulation in mouse lungs [197], and is universally correlated with ARDS, AKI, morality [198], and the need for mechanical ventilation in COVID-19 [197]. IL-13 is also known to upregulate monocyte-macrophage-derived suppressor cells (MDSC), which play a role in immune suppression and lymphopenia, a hallmark of severe COVID-19 disease [199,200,201,202].

PGD_2_/DP2 receptor signaling is thought to play a role in fibrosis. PGD_2_/DP2 receptor signaling exerts direct pro-apoptotic and pro-fibrotic actions on various cells, including islet cells, cardiomyocytes, and osteoclasts [203,204,205]. Moreover, PGD_2_/DP2 receptor-mediated IL-13 release is a significant inducer of fibrosis by stimulating the IL-13Rα_2_ receptor expressed on macrophages to release TGF-β1 [206]. Therefore, elevated levels of TxA_2_ in COVID-19 and SCD may be rapidly converted to 11dhTxB_2_ in the lungs, thereby leading to DP2 receptor signaling and fibrosis in the lungs and heart.

## 8. Therapeutic Options for Thrombo-inflammation in COVID-19 and SCD: Past, Present, and Future

Blood transfusion is the only treatment for SCD-associated vaso-occlusive crises or thrombotic events, such as acute painful episodes, cerebral infarction, and acute chest syndrome [207]. Widely varying anticoagulants, including heparin and its analogs, are used in both arterial and venous thrombosis associated with SCD, but they have proven ineffective in preventing acute pain episodes [208,209]. As a treatment of acute pain crisis in SCD, tinzaparin, low molecular weight heparin, was shown to reduce the severity and crisis duration in a double-blind, randomized, controlled trial [210].

Antiplatelet agents, such as aspirin, prasugrel, and ticagrelor, have also been tested in SCD (Table 3). In a double-blind crossover study of children with SCD, low-dose aspirin, an inhibitor of COX-1 action, did not affect the frequency and severity of vaso-occlusive crises, compared to placebo [211]. Prasugrel and ticagrelor are P_2_Y_12_ receptor antagonists that block platelet stimulation induced by ADP. In a double-blind, randomized, placebo-controlled trial of prasugrel in children and adolescents with SCD, prasugrel was found to be safe, but did not reduce the rate of vaso-occlusive crisis or diary-reported events over a 9–24-month period [212]. Ticagrelor demonstrated no effect on diary-reported pain in young adults with SCD [213]. However, as discussed above, heme-driven CLEC2-induced platelet activation is dependent on ADP stimulation of platelet P_2_Y_1_ receptor, but not the P_2_Y_12_ receptor [84], while prasugrel only blocks the latter [214]. The failure of prasugrel and ticagrelor may also indicate that other P_2_Y_12_ antagonists may not be effective in SCD [213,214]. Unfortunately, there are no approved P_2_Y_1_ antagonists to our knowledge. Moreover, other antiplatelet therapies have been tested in clinical trials, including eptifibatide, a platelet αIIbβ3 receptor blocker, which failed to improve time to crisis resolution or hospital discharge in SCD patients, though only 13 patients were enrolled in the study [215]. Meanwhile, targeting IL-1β downstream of inflammasome activation with canakinumab led to improved thrombo-inflammation and quality of life, including reduced days of hospitalization and pain [216]. P-selectin antibody (Crizanlizumab) is the only targeted therapy against thrombo-inflammation in SCD. However, prophylactic P-selectin inhibition by crizanlizumab led to only a ~50% reduction in hospitalization related to VOC, suggesting that P-selectin-independent pathways contribute to the remaining morbidity of VOC [85].

Therapies targeting thrombo-inflammation in patients with COVID-19 are in clinical trials [217]. In a necropsy study of 68 COVID-19 patients, nearly 70% (48 out of 68) were treated with anticoagulants, and of those treated with anticoagulants, almost 50% had large thrombi, and nearly 90% had microvascular thrombi of arterioles and capillaries (42 out of 48). This demonstrates the lack of efficacy of anticoagulation in severe COVID-19 [218]. Furthermore, prohibitive signal for bleeding risk, in addition to futility, has recently led to the discontinuation of high-dose heparin arm in REMAP-CAP, ACTIV-4, and ATTACC studies in severe COVID-19 [218,219,220]. The momentum seems to be finally shifting to antiplatelet agents, both for prevention and treatment, even though the efficacy of these or other agents remains to be demonstrated in preclinical models of COVID-19.

Numerous clinical trials have been initiated to investigate the benefits of antiplatelet therapy in COVID-19. In the NIH ACTIV-4 trial, amongst 657 symptomatic outpatients with COVID-19, the major adverse cardiovascular or pulmonary outcomes were not significantly different for patients randomized to low-dose aspirin, apixaban (2.5 mg twice daily), apixaban (5.0 mg twice daily), or placebo [221]. However, the median time from diagnosis to randomization and from randomization to initiation of study treatment were 7 days and 3 days, respectively, suggesting that survival bias may account for a very low event rate in both placebo and treatment groups. Early administration of aspirin in ambulatory COVID-19 patients appears to provide significant benefit and correlates with a decrease in overall mortality [222], but was not effective in reducing progression to invasive mechanical ventilation or death in hospitalized patients, as reported in the Randomized Evaluation of COVID-19 thERapY (RECOVERY) trial, the world’s largest clinical trial of treatments for patients hospitalized with COVID-19 [223,224]. Aspirin irreversibly inhibits both COX enzymes (COX-1 >> COX-2), preventing prostaglandin production by cells until the new enzyme is produced [225]. Low doses of aspirin, typically 75 to 81 mg/day, are sufficient to irreversibly acetylate serine 530 of COX-1, but have little effect on COX-2 [225]. Therefore, the lack of efficacy of aspirin in hospitalized patients with COVID-19 may be due to inducible COX-2-mediated TxA_2_ generation and the failure of aspirin to inhibit the effects of TxA_2_ once synthesized. Moreover, the well-known phenomenon of aspirin resistance in the obese or the elderly has been attributed to increased expression of cytosolic phospholipase A_2,_ and COX-2, which leads to increased generation of TxA_2_ [226]. Marked increase in TxA_2_ generation and COX-2 expression in severe COVID-19 raises the specter of aspirin resistance, especially in the elderly or obese patients, as in the general population [227,228]. Plasma from patients with COVID-19 triggered platelet and neutrophil activation and NET formation in vitro; the latter was blocked by therapeutic-dose low-molecular-weight heparin, but not by aspirin [229]. Moreover, the use of aspirin in children with COVID-19 is relatively contraindicated due to the risk of Reye’s syndrome [230].

Furthermore, COX-2 inhibitors are known to increase the risk of cardiovascular events and, therefore, are not advised in SCD or COVID-19 [231,232]. Blocking COX-1 or COX-2 may result in more challenges than cures because of their broad inhibition of several essential prostanoids other than TxA_2_ [233]. Although TxA_2_ synthase inhibitors suppress TxA_2_ formation, accumulation of the substrate prostaglandin (PG)H_2_ stimulates TP receptor on platelets and the endothelium, thereby inhibiting the antiplatelet action of TxA_2_ synthase inhibitors [234]. TP receptor antagonists block the activity of both TxA_2_ and PGH_2_ on platelets and the endothelium [234]. Thus, early administration of well-tolerated TP receptor antagonists may limit progression to severe COVID-19 [81] and may also be effective in SCD, considering the common pathobiology of thrombo-inflammation in the two disease states.

It has been proposed that blocking the deleterious effects of PGD_2_ and TxA_2_ with a dual DP2/TP receptor antagonist, ramatroban, might be beneficial in COVID-19 [136,235]. Ramatroban is a surmountable and potent antagonist of TP receptors. Ramatroban is orally bioavailable and has been used in Japan for the past 20 years as a treatment for allergic rhinitis. Ramatroban has been shown to provide rapid relief of symptoms and successfully treat four patients with severe COVID-19 [168]. This is consistent with the role of TP receptor antagonism in relieving postcapillary pressures, promoting resolution of edema, and improving lung compliance and gas exchange [164,168]. Notably, patients treated with ramatroban did not develop overt long-haul COVID symptoms after recovery from acute illness, supporting the anti-fibrotic effect of ramatroban, as demonstrated in a mouse model of silicosis [168,236].

Ramatroban is 100-150 times more potent than aspirin in inhibiting platelet aggregation, P-selectin expression, and sphingosine-1-phosphate (S1P) release from platelets [237,238,239]. S1P is chemotactic for monocytes and inhibition of S1P release reduces monocyte infiltration [239]. Ramatroban also decreases macrophage infiltration by inhibiting endothelial surface expression of ICAM-1 and VCAM-1, and inhibiting MCP-1 expression on endothelial cells in response to TNF-α or platelet-activating factor [237]. Additionally, the potentiation of CLEC2 signaling by TxA_2_ was abolished by 10 µM ramatroban, while 1 mM aspirin was only partially effective [84]. In addition to its anti-platelet action, ramatroban also improves vascular responsiveness [237]. With a plasma half-life of about 2 h, the antiplatelet action of ramatroban is reversible [237]. This is of advantage in the event of bleeding complications following anticoagulation in critically ill COVID-19 patients [240], and SCD patients with venous thromboembolism [241].

## 9. Conclusions

Thrombo-inflammation is a classic feature of both SCD-associated vaso-occlusive crisis and severe COVID-19. Thrombo-inflammation leads to microvascular thrombosis and thrombotic microangiopathy. Both diseases share the pathobiology of endothelial cell injury and activation, platelet activation, and platelet-leukocyte partnership, culminating in thrombo-inflammation. COX-2-mediated increase in TxA_2_ signaling may play an important role in platelet activation, leading to thrombo-inflammation in both SCD and COVID-19. The failure of the previous anticoagulant and antiplatelet strategies in both SCD and COVID-19 underlines the importance of identifying new therapeutic targets, such as TxA_2_ and/or PGD_2_, for the resolution of acute crisis in both diseases.

## Figures and Tables

**Figure 1 biomedicines-11-00338-f001:**
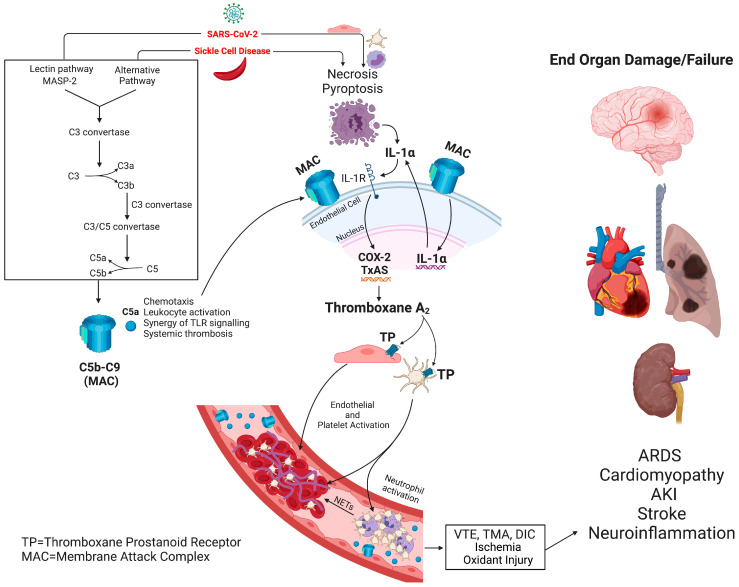
Putative mechanism of complement-mediated microvascular thrombosis and vaso-occlusive disease in SCD and COVID-19: SARS-CoV-2 infection and sickle cell disease induce complement activation and formation of membrane attack complex leading to necrosis and pyroptosis of endothelial cells, platelets, and monocytes and accumulation of IL-1α. IL-1α stimulates the IL-1 receptor expressed on endothelial cells leading to thromboxane synthesis. Thromboxane A_2_ via the TP receptor activates platelets leading to platelet activation, platelet neutrophil partnership, neutrophil activation, and the release of neutrophil extracellular traps (NETs), thrombo-inflammation, oxidative stress, and subsequent end-organ damage and failure. The current review article is primarily focused on eicosanoid signaling in platelets; therefore, other receptors and pathways were excluded from Figure 1 for the reader’s convenience. COX, cyclooxygenase; IL, interleukin; NETs, neutrophil extracellular traps; TP, thromboxane prostanoid receptor; MAC, membrane attack complex; VTE, venous thromboembolism; TMA, thrombotic microangiopathy; DIC, disseminated intravascular thrombosis; ARDS, acute respiratory distress syndrome; AKI, acute kidney injury.

**Figure 2 biomedicines-11-00338-f002:**
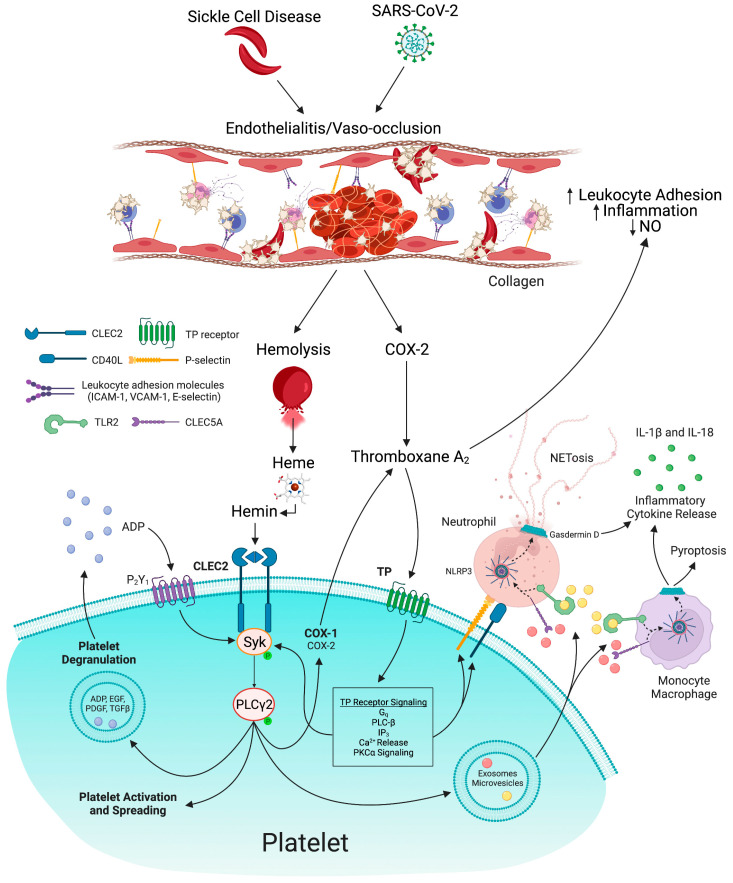
Mechanisms of heme and thromboxane A_2_-mediated thrombo-inflammation in COVID-19 and sickle cell disease (SCD): Vaso-occlusion due to sickling or direct entry by SARS-CoV-2 virus leads to endothelial cell activation and damage, and hemolysis. COX-2 expression in endothelial cells promotes thromboxane A_2_ synthesis. Thromboxane A_2_ inhibits nitric oxide (NO) synthesis and promotes leukocyte adhesion and thrombo-inflammation. Free heme released from red blood cells is spontaneously oxidized to its ferric form, hemin. Hemin stimulates platelet CLEC2 signaling and thromboxane A_2_/TP receptor-dependent Syk phosphorylation leading to platelet activation, spreading, and degranulation. Platelets release exosomes and microvesicles, which stimulate the CLEC5A and TLR2 receptors on neutrophils. Subsequently, NLRP3 activation in neutrophils and monocytes promotes activation and assembly of gasdermin D, leading to the release of neutrophil extracellular traps and monocyte pyroptosis. NLRP3 inflammasome activation induces the release of proinflammatory cytokines, including IL-18 and IL-1β, thereby fueling thrombo-inflammation in COVID-19 and SCD. NO, nitric oxide; COX, cyclooxygenase; IL, interleukin; NETs, neutrophil extracellular traps; TP, thromboxane prostanoid receptor; CLEC, C-type lectin-like receptor; Syk, spleen tyrosine kinase; PLC, phospholipase C; PKC, protein kinase C; TLR, toll-like receptor; ADP, adenosine diphosphate; EGF, epidermal growth factor; PDGF, platelet-derived growth factor; TGF, transforming growth factor; NLRP3, NLR family pyrin domain containing 3.

**Table 1 biomedicines-11-00338-t001:** Comparative analysis of Plasma D-dimer levels in SCD during the steady state and sickle crisis and in COVID-19 patients.

Subjects	Age/Reference	Plasma D-Dimer Levels		*p*-Value Compared to Controls
		Control	Disease State	
Sickle Cell Disease	Adult [76]	HD (*n* = 35) 79 ± 25 ngh	Steady State SCD *n* = 25 (Samples = 28) 566 ± 739 ng/mL	*p* < 0.001
			SCD Painful Crisis *n* = 21 (Samples = 40) 1038 ± 1010 ng/mL	*p* < 0.001
	12–37 years [75]	SCD with pain crisis and normal chest X-ray (*n* episodes = 32) 584.2 µg/L (250–3119 µg/L)	SCD with pain crisis and abnormal chest X-ray (*n* episodes = 13) 2117.0 µg/L (250–9143 µg/L)	N/A
	Unventilated: 62.5 ± 8.4 Ventilated: 53.8 ± 9.3 [77]	Hospitalized COVID-19 patients did not require artificial ventilation (*n* = 18) 650 ± 175 ng/mL	Hospitalized COVID-19 patients requiring artificial ventilation (*n* = 11) 1250 ± 210 ng/mL	*p* < 0.05
COVID-19	65.57 ± 13 years [78]	COVID-19 patients without pulmonary embolism (*n* = 118) 1310 ng/mL (800–2335)	COVID-19 patients with pulmonary embolism (*n* = 44) 5364 ng/mL (2928–12,275)	*p* = 0.001

**Table 2 biomedicines-11-00338-t002:** Comparative analysis of thromboxane levels in SCD, COVID-19, and asthmatics.

Subjects	Source and Analyte	Thromboxane Levels		*p*-Value Compared to Controls
		Control	Disease State	
Sickle Cell Disease	Plasma 2,3 dinor-TxB_2_ [140] (µg/L) (Mean ± SEM)	HD (*n* = 12) 2.75 ± 0.83	Steady State SCD (*n* = 15) 21.53 ± 5.10	*p* < 0.001
	Plasma TxB_2_ [140] (µg/L) (Mean ± SEM)	HD (*n* = 12) <0.005	Steady State SCD (*n* = 15) 0.543 ± 0.101	*p* < 0.05
	Urinary TxB_2_ [140] (pg/mg creatinine) (Mean ± SEM)	HD (*n* = 12) 0.41 ± 0.30	Steady State SCD (*n* = 15) 0.91 ± 0.13	*p* < 0.05
	Urinary 2,3 dinor-TxB_2_ [140] (pg/mg creatinine) (mean ± SEM)	HD (*n* = 12) 1.70 ± 0.032	Steady State SCD (*n* = 15) 2.81 ± 0.13	*p* < 0.01
	Urinary 11-dehydro-TxB_2_ [141] (pg/mg creatinine) (Mean ± SEM)	HD (*n* = 33) 299 ± 20	Steady State SCD (*n* = 49) 1227 ± 191	*p* = 0.0002
			Vaso-Occlusive SCD (*n* = 15) 1836 ± 536	*p* = 0.0005
COVID-19	BALF TxB_2_ [52] (nmol/L) (Means)	HD (*n* = 25) <0.250	Severe COVID-19 (*n* = 33) 12.0	*p* < 0.0001
	Plasma TxB_2_ [23] (ng/mL) (Median)	HD (*n* = 11) 4.0	Severe COVID-19 (*n* = 35) 7.5	*p* < 0.05
	Urinary 11-dehydro-TxB_2_ [143] (pg/mg creatinine) (Median (95% CI))	Without Events (*n* = 47) 4890 (5049–8290)	With Events (*n* = 18) 7603 (7541–19,791)	*p* = 0.002
		<10 d of hospitalization (*n* = 35) 4801 (3817–9196)	≥10 d of hospitalization (*n* = 30) 8614 (7990–14,316)	*p* = 0.02
		No death (*n* = 48) 5360 (5907–10,038)	Death (*n* = 6) 15,069 (1915–42,007)	*p* = 0.004
		No Mechanical Ventilation (*n* = 56) 5137 (4498–7512)	Mechanical Ventilation (*n* = 9) 20,121 (5364–41,015)	*p* < 0.001
Atopic Asthmatics	BALF TxB_2_ [144] (nmol/L) (Mean ± SEM)	Before Allergen Challenge (*n* = 8) 0.130 ± 0.021	After Allergen Challenge (*n* = 8) 0.430 ± 0.108	*p* < 0.05

HD, healthy donor; BALF, bronchoalveolar lavage fluid.

**Table 3 biomedicines-11-00338-t003:** Efficacy trials of antiplatelet therapies in SCD.

Study Design	Study Population	Intervention	Primary Outcome Measure and Result
Phase IIb Multicenter, double-blind, double-dummy, randomized, placebo-controlled, parallel-group [213]	● Have SCD [homozygous sickle cell (HbSS) or sickle beta-zero thalassemia (HbSβ^0^)] ● Ages 18–30 years (mean 22.2 years old) ● Have ≥4 days of pain during the 4-week single-blind placebo baseline period prior to randomization ● If on hydroxycarbamide, a stable dose for 3 months prior to enrollment required ● If on erythropoietin, drug must have been prescribed 6 months before and at a stable dose for ≥3 months prior to randomization (*n* = 194)	● Ticagrelor 10 mg plus matching placebo for ticagrelor 45 mg ● Ticagrelor 45 mg plus matching placebo for ticagrelor 10 mg ● Matching placebo for ticagrelor 10 and 45 mg Duration: 12 weeks	Proportions of days with diary-reported SCD pain No significant difference between placebo and ticagrelor treatment groups
Phase III Multinational, double-blind, randomized, placebo-controlled, parallel-group [212]	● Have SCD [homozygous sickle cell (HbSS) or sickle beta-zero thalassemia (HbSβ^0^)] ● Are participants with SCD who have had ≥2 episodes of vaso-occlusive crisis (VOC) in the past year ● Have a body weight ≥ 19 kilogram (kg) and are ≥ 2 and <18 years of age, inclusive at the time of screening ● If participants are ≥2 and ≤16 years of age, they must have had a transcranial Doppler within the last year (*n* = 341)	● Prasugrel 0.08–0.12 mg/kg po once daily ● Placebo Duration: 9–24 months	Number of Vaso-Occlusive Crisis (VOC) Events Per Participant Per Year (Rate of VOC) Terminated due to lack of efficacy
Phase III Double-blind crossover study [211]	● Have sickle hemoglobinopathy observed regularly ● Ages 2–17 years old (mean 7.7 years old) ● The hematologic diagnosis was confirmed by cellulose acetate electrophoresis at pH 8.6 and citrate agar electrophoresis at pH 6.4 ● At least 50% compliant (*n* = 49)	● Low dose aspirin ● Placebo	Frequency and severity of VOC No significant difference between placebo and aspirin treatment groups

## Data Availability

Not applicable.

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
