# Peer review of "Thrombo-Inflammation in COVID-19 and Sickle Cell Disease: Two Faces of the Same Coin"

_biomedicines, 2023, doi:10.3390/biomedicines11020338_

Round 1

Reviewer 1 Report

An interesting review.

Accept as it is

Author Response

We are most appreciative of the comments from Reviewer 1 which are most encouraging. 

Reviewer 2 Report

The review is a general overlook of platelet function in sickle cell disease. The review, in general, is well written; however, some issues weren't covered in the review. Figure 1 is partially informative. There is no ADP/ATP ratio involved in platelet function there is no involvement of coagulation receptors in the figure. The article 10.26574/maedica.2020.16.2.268 has not been cited. Figure 2 has the same problem; there is no complete analysis of the process, only one receptor is involved. Why PLA2 and Lipooxygenase enzymes are not described? In addition, cAMP induced by PGE2 is not observed.

The text requires more context on the role of heparin or anticoagulation factors and how the consumption of coagulation proteins may be crucial in the process.

The conclusions should be rewritten. 

Author Response

We are most appreciative of the comments and helpful suggestions from Reviewer 2. Please see below a point-by-point response. 

Comment from Reviewer 2: The review is a general overlook of platelet function in sickle cell disease. The review, in general, is well written; however, some issues weren't covered in the review. Figure 1 is partially informative. There is no ADP/ATP ratio involved in platelet function there is no involvement of coagulation receptors in the figure.

Response from authors: The figure is focused on the role of eicosanoids. Adding numerous other factors will make it unwieldy while risking loss of focus. This has been clarified in the legend of figure 1.

Comment from Reviewer 2: The article 10.26574/maedica.2020.16.2.268 has not been cited.

Response from authors: The suggested article has been cited on page 4.

Comment from Reviewer 2: Figure 2 has the same problem; there is no complete analysis of the process, only one receptor is involved. Why PLA2 and Lipooxygenase enzymes are not described? In addition, cAMP induced by PGE2 is not observed.

Response from authors: The role of the lipoxygenase pathway in thromboinflammation remains poorly defined compared to the role of thromboxane A2 and hence the focus of the figure. In the text, we have added reference to the lipoxygenase pathway in both SCD and COVID (please see page 13).

Reviewer 3 Report

The article entitled “Thrombo-inflammation in COVID-19 and Sickle Cell Disease: Two Faces of the Same Coin” is a review focused on the similarity of two diseases such as COVID-19 and Sickle Cell Disease in terms of pathogenesis and clinical characteristics. The authors made an excellent editorial work. This review is exhaustive and it is full of detailed biological, clinical and laboratory  informations. The writing is smooth and flowing and the reading is simple and understandable. The idea of the authors is original and I think that it is important to spread this article to the scientific community. I have no any comments. Therefore, I think that this article is suitable for publication in its current version.

Author Response

We are most appreciative of the comments from Reviewer 3 which are most encouraging. 

Round 2

Reviewer 2 Report

The manuscript was improved with the modifications.